# Acid and Rennet Coagulation Properties of A2 Milk

**DOI:** 10.3390/foods11223648

**Published:** 2022-11-15

**Authors:** Bibiana Juan, Antonio-José Trujillo

**Affiliations:** Centre d’Innovació, Recerca i Transferència en Tecnologia dels Aliments (CIRTTA), XIA, TECNIO, Departament de Ciència Animal i dels Aliments, Facultat de Veterinària, Universitat Autònoma de Barcelona (Cerdanyola del Vallès), 08193 Barcelona, Spain

**Keywords:** β-casein, genetic polymorphism, A2 milk, acid and rennet coagulation

## Abstract

This study investigated the acid and rennet milk coagulation properties of A2 milk (β-casein (CN) A2A2 genotype), in comparison to a control milk (blend of A2A1/A1A1/A2A2 genotypes). Acid and rennet coagulation were evaluated using the Optigraph^®^ system, measuring the coagulation time, aggregation rate, and gel density or curd firmness. The acidification kinetics were monitored using a CINAC^®^ system, evaluating the time to reach pH 4.6, the acidification rate, the maximum acidification rate, the time required to reach it, and the latency time. The water-holding capacity of acid milk gels and the potential yield, total solids, and syneresis of enzymatic gels were also evaluated. Some variables were highly influenced by the farm factor, showing the importance of the effect of extrinsic parameters. Acid and enzymatic coagulation times were not affected in either milk. The A2 milk presented higher acid gel density and latency time than the control milk. Although the differences in water-holding capacity were not statistically significant, the A2 milk presented lower values, related with the higher gel density. The A2 milk also showed higher rennet aggregation rate and curd firmness than the control milk. Potential yield and syneresis were higher in the A2 milk, which is in accordance with the higher firmness of curd. Coagulation results and gel and curd properties indicate that it is possible to manufacture acid and rennet coagulation dairy products from A2 milk with no major differences when compared with a control milk.

## 1. Introduction

Cow’s milk contains an average of 3.5% of protein, of which 80% is comprised of four caseins (α_S1_-CN, α_S2_-CN, β-CN and κ-CN) and 20% is comprised of whey proteins (mainly α-LA and β-LG). β-CN accounts for approximately 30% of total milk protein, and in bovine milk, diverse mutations happened leading to 13 genetic variants (A1, A2, A3, B, C, D, E, F, G, H1, H2, I and J) [1], while a fourteenth allele, A4, has been detected in Korean native cattle [2]. A1 and A2 variants are the most common and distinguished in regards to the presence of proline or histidine at the 67th position of the casein [3]. This single amino acid difference results in the alteration of protein functions and functionalities, enzymatic and acidic hydrolysis, and liberation of bioactive peptides that could have an influence on further milk processing and human nutrition [4]. Some studies have indicated that β-CN A1 digestion releases a greater quantity of beta-casomorphine-7 (BCM-7) than β-CN A2 [5,6,7,8], which is considered to be a health risk as it can potentially affect opioid receptor in the nervous, endocrine and immune system [9,10], and is associated with the development of some non-transmissible diseases [11,12]. However, the European Food Safety Authority (EFSA) did not find any clear evidence of a cause-effect relationship between BCM-7 and the development of certain non-transmissible disease, and only stated that BCMs may cause disturbances of the gastrointestinal tract, such as swelling, flatulence and abdominal pain [13]. Reviews carried out after EFSA’s report also identified a potential link between the consumption of β-CN A1 and its derivative BCM-7 with an increase in the inflammatory response and discomfort at the intestinal level [14,15,16], concluding that β-CN A2 can have some beneficial effects on the gastrointestinal system [17]; meanwhile, it is not confirmed that β-CN A1 has negative effects on human health [18]. Although further investigation about the health effects of β-CN A2 is necessary, [15,16,18,19], this milk continues to gain prominence on the market. Thus, further research is required to understand the techno-functional properties of this milk.

β-CN genotype A2A2 is the most frequent genotype in Estonian cattle [20], Danish Jersey cows [4,21], and Norwegian Red cows [22], followed by its heterozygous genotype A1A2, while A1A1 and the rest are rare [22,23]. Cows with A2A2 genotype produce milk containing β-CN A2, unlike cows with A1A1 and A1A2 genotypes, who synthesize β-CN A1 [24]. Currently in Spain and other European countries, most commercially available milk contains a mixture of β-CN A1 and A2; however, a growing trend towards selecting A2 only is taking place.

Genetic milk proteins have an effect in dairy processing, particularly in coagulation properties necessary for yoghurt and cheese making [25,26,27]. However, there are only few reports on the influence of the β-CN genotype on dairy product manufacturing.

Jensen et al. (2012) and Poulsen et al. (2013) associated β-CN A2 with poorly coagulating milk [4,28]. Hallén et al. (2007) [29] described that the A2A2 genotype of β-CN had a negative effect on rennet-induced gelation, observing shorter clotting time and higher gel firmness in milk containing β-CN A1A2 when compared with A2A2 [21,22]. However, other authors did not find any effect of the β-CN alleles on rennet coagulation properties [23,30,31,32]. Similarly, the β-CN genotype was not found to have any effect on acid-induced gel strength or acid coagulation time [26]. Nguyen et al. (2018) described longer acid gelation time and lower gel strength firmness in milk containing β-CN A2A2 than milk containing A1A1 [33].

The objective of the present work was to study the acid and enzymatic coagulation properties of tank milk obtained from cows with the β-CN A2A2 genotype, in comparison to a control milk obtained from cows with A1A2, A1A1 and A2A2 β-CN genotypes.

## 2. Materials and Methods

### 2.1. Sample Collection

This experiment used A2 tank milk (cows with β-CN A2A2 genotype) and control tank milk (cows with β-CN A2A1, A1A1 and A2A2 genotypes) obtained from three different farms on two occasions from April to July 2021. The cows (total 452) were selected based on casein genotypes, days in lactation, and number of lactations, and they were divided into two groups on each farm (A2 and control). Genetic information relating to individual cows was obtained from FEFRIC (Federació Frisona de Catalunya) and can be seen in Table 1 and Table 2.

All tank milks from the morning milking were collected and cooled before being transported to Universitat Autònoma de Barcelona. Once at the laboratory, milks were tempered at 50 °C to be skimmed, and immediately cooled and stored in a cold chamber (4 °C) before being subjected to analysis.

### 2.2. Milk Composition

Milk samples were analyzed for total solids [34] and pH with a pH meter (Crison Micro-pH 2001, Crison Instruments, Barcelona, Spain). The total nitrogen was analyzed using the Kjeldahl method [35], and protein content was calculated by multiplying the total nitrogen content by 6.38.

### 2.3. Evaluation of Acid Coagulation Properties

#### 2.3.1. Preparation of Samples

Before fermentation, the milk was heated at 80 °C for 20 min in order to eliminate spoilage microorganisms and possible pathogens present in the milk, and to improve the viscoelastic properties of the product. This method is commonly used in the industry to create dairy products. The heat treatment causes a denaturation of whey proteins, especially β-LG that associates with κ-CN on the surface of casein micelles [36].

#### 2.3.2. Preparation of Starter Culture

For acid coagulation, a commercial lyophilized culture of *Lactobacillus delbruekii* subsp. *bulgaricus* and *Streptococcus salivarius* subsp. *thermophilus* (YO-MIX 300 LYO 10 DCU, DANISCO FRANCE SAS, Dangé-Saint-Romain, France), in the proportion of 1:1 was used. Skim milk was heated to a temperature around 43 °C, and 0.2 g of the starter culture was added, mixed, and incubated at 43 °C until it reached a pH of around 4.8.

#### 2.3.3. Evaluation of Coagulation Parameters

Skim milk samples were pre-heated in a water bath until they reached the temperature of 43 °C, and subsequently were acidified with 2% of the starter culture.

Acid coagulation was monitored using the Optigraph^®^ system (Ysebaert, Frépillon, France), which is based on the measurement of NIR signal attenuation. This device passes an infrared beam through a sampling tube containing milk. A sensor on the other side measures the amount of light absorbed by the milk as it coagulates, and the changes are analyzed in real time by a computer that converts them into directly usable data. The cuvettes were each filled with 10 mL of inoculated milk and monitored for 4 h at 43 ± 2 °C. Both samples of milk were analyzed in quadruplicate. From the coagulation curves, three parameters were obtained: time for detecting the onset of coagulation (CT), which is indicated by the maximum of the first derivate curve; aggregation rate (AR), calculated from the slope of the linear region of the curve; and gel density index (GD), calculated as the differences between D1 and D0 [37].

The acidification kinetics were monitored using a CINAC^®^ system (Ysebaert, Frépillon, France), which allows the follow-up in continuous mode of the acidifying activity of lactic starters. Each type of milk was fermented in three replicated flasks with 50 mL of volume and monitored until it reached a pH of 4.6. The parameters assessed from the acidification curves process were the time of fermentation (*T*_f_), which is the time to reach a pH of 4.6; the mean of acidification rate (*V*_*m*_), defined as the slope of the straight section of the curve; the maximum acidification rate (*V*_*max*_), which corresponds to the inflection point of the pH curve versus time; the time required to reach *V*_*max*_, being the time at which the maximum acidification rate was observed (*T*_*max*_); and the latency time (*T*_*a*_), which is the time necessary for a pH decrease of 0.08 units to be produced.

#### 2.3.4. Water-Holding Capacity of Gels

Samples of 200 mL of each type of skim milk was heated to 43 °C and then inoculated with the starter culture at 2%. Inoculated milk (40 g) was distributed in three centrifuge tubes for each sample and placed to incubate at 43 °C for 4 h. Subsequently, the tubes were placed in a cold chamber at 4 °C for 24 h. Before centrifugation, coagulated milks were warmed at room temperature and afterwards centrifuged (Sigma 4K15, Postfach, Germany) at 5000× *g * for 20 min at 20 °C. Succeeding centrifugation, whey was weighed, and water-holding capacity (WHC) results were expressed as grams of expelled whey per 100 g of milk.

### 2.4. Evaluation of Rennet Coagulation Properties

Raw skim milk was warmed to 32 °C, and rennet (chymosin 80%, with a declared activity of 1:10,000, Laboratorios Arroyo, Santander, Spain) was added at 0.03% (*v*/*v*). Coagulation was carried out at 32 °C for 30 min. Rennet coagulation time (RCT), rate of curd firming (RCF), and curd firmness at 30 min (CF_30_) were assessed in quintuple by the Optigraph^®^ system.

#### 2.4.1. Potential Yield and Total Solids of Curds

Milk samples (200 mL) were warmed to 32 °C and rennet was added at 0.03% (*v*/*v*). Portions of the renneted milks (40 g) were transferred into centrifuge tubes and allowed to coagulate at 32 °C for 30 min. The coagulum was centrifuged (Sigma 4K15) at 13.000× *g* for 15 min at 10 °C. Potential yield of curds, expressed as grams of retained curd per 100 g of milk, was determined by weighing the obtained pellets.

Curds were analyzed in triplicate for total solids content by drying in an oven at 102 °C [38].

#### 2.4.2. Rennet Curd Syneresis

Curd syneresis was evaluated according to the method of Marshall (1982) [39]. Skim milk (50 mL) was coagulated at 32° C for 30 min by the addition of 0.03% of rennet. After coagulation, the curd was cut crosswise and around the sides of the beaker with a spatula and left to stand at room temperature. After one hour, a plastic grid was placed on the curd surface. This retained the curd while allowing the whey to be poured off. Syneresis was measured by weighing the quantity of whey removed.

### 2.5. Statistical Analysis

Data were processed by multifactor analysis of variance (ANOVA) using the randomized block design of R Commander [40], considering both milk types’ (A2 and control) farm and their interaction as factors. Tukey’s range test was used for comparing the means of the milks, and evaluations were based on a significance level of *p* ≤ 0.05. Data relating to milk composition were compared by Student’s *t*-test for independent samples. The complete experiment was repeated on two independent occasions on each of the three farms.

## 3. Results and Discussion

### 3.1. Milk Samples

The cows used in the present study were selected based on their days in lactation and number of lactations in an attempt to make each group as similar as possible (Table 1).

The control tank milk was a mixture of milks from animals genotyped for β-CN with mostly A1A2, followed by A1A1 and a small amount of A2A2 (Table 2). Three β-LG variants were found, with similar distribution in both milks, showing a tendency in the following order: AB > AA > BB. As regards κ-CN, in the A2 milk, only three κ-CN variants were found (AA, AB and BB). However, in the control milk, we also found BE, AE and EE, in accordance with Hallén et al. [26] in Swedish Red and Holstein breeds, and Vallas et al. [41] in Estonian Holstein cows. Comin et al. [23] mentioned that the κ-CN E allele rarely occurred in association with β-CN A2A2. Hallén et al. [26] described that the rare κ-CN genotype EE was associated only with the β-CN A1A1 genotype, while genotypes BE and AE of κ-CN were also mainly associated with β-CN genotypes that included A1. In both milks, the allele AB presented in higher frequency.

### 3.2. Milk Composition

The chemical composition of the milk samples are shown in Table 3.

The skim milk samples presented an average value of 6.70 ± 0.06, 3.13 ± 0.15 (%), and 9.14 ± 0.20 (%) for pH, protein, and total solids-non-fat, respectively. These results were in line with other studies on cows’ milk composition [21,30,42,43]. No statistical differences were observed in any parameter between the A2 and control milks, consistent with Nguyen et al. [33], who did not find significant differences in the concentration of fat, protein, and total solids between bovine milk with two different β-CN phenotypes (A1 and A2).

A significant farm effect was observed on the percentage of protein (*p* ≤ 0.05). Likewise, no differences were observed between the A2 and control tank milks.

### 3.3. Acid Coagulation Properties

Table 4 shows the acid coagulation properties of milks. No statistical differences between the two types of milks were found in clotting time. These results align with those found by Ketto et al. [22], who noticed that β-CN polymorphism did not affect the acid coagulation properties of milk, showing similar gelation time in β-CN A2A2 milk and A1A2, measured by Formagraph. Hallén et al. [26] also described that there was no significant effect of the β-CN genotype on acid coagulation in Swedish Red and Swedish Holstein breeds, while the β-LG genotype significantly influenced the acid coagulation time, largely due to its impact on β-LG concentration.

A significant farm effect and their interaction with the genotype was observed for aggregation rate (*p* ≤ 0.05) (Table 3), but no statistical differences were observed between the A2 and control tank milks in global results.

The milk type influenced the acid gel density characteristics (Table 4), with A2 milks presenting higher GD than control milks. Our results disagree with those of Nguyen et al. [33], who found that yoghurts produced with β-CN A2A2 milk presented lower storage modulus and gel firmness than those produced with A1A1 milk, indicating a softer gel. Unfortunately, in the study of Nguyen et al. [33], the genetic polymorphism of the other milk proteins were not facilitated and taken into consideration, which could be the reason for the differences observed in gel density. Hallén et al. [26] and Ketto et al. [22] described that the individual β-CN polymorphism did not affect the acid coagulation properties of milk. However, the different composite genotypes were found to have a significant effect on gel firmness and gel firming rate, showing higher values with the composite α_s1_-β-κ-CN BB-A2A2-AA compared to BB-A2A1-AA, revealing that for the same genotype as it relates to α_s1_-CN and κ-CN, milks with β-CN A2A2 have greater gel firmness than A1A2 milks, in agreeance with our results [22].

The results for the acidification kinetics performed with a CINAC^®^ device are shown in Table 5.

No statistical differences were found in global results for maximum acidification rate (*V_max_*), mean acidification rate (*V_m_*), and time to reach maximum acidification rate (*T_max_*), showing a medium value of 0.0196 pHU/min, 0.0187 pHU/min, and 109.15 min, respectively. These results are consistent with those of Nguyen et al. [33], who described no significant differences in the *V_max_* between β-CN A2A2 and A1A1 milks. These authors did not find differences in the fermentation curve of either of these milks, coinciding with our results (Figure 1).

Fermentation time ($T_{f})$, defined as the time for the milk to reach pH 4.6, was approximately 188 min, with no statistical differences between A2 and control milks (Table 5). However, a significant effect of farm was observed (*p* ≤ 0.01). Despite not showing any statistical differences, we observed the same trend in all farms: A2 content was higher than in the control milk. Nguyen et al. (2018) also found no significant differences in the fermentation time between β-CN A2A2 and A1A1 milks [33].

Latency time (*T_a_*) was highly influenced by milk type (*p* ≤ 0.001), being longer in the A2 milk than the control milk (Table 5). This faster acidification rate at the beginning of fermentation in the control milk aligns with the higher *V_m_*, $\;V_{mar}\;$and shorter $\;T_{e},\;$ and $\;T_{m}\;$of the control milk, although differences in these parameters were not statistically significant (Table 5).

#### Water Holding Capacity of Acid Gels

The WHC, represented as the quantity of whey expulsion after a forced centrifugation process, is an estimation of the water-retention capacity of gel. No significant differences were found in the global results of WHC between milks, with values of 81.77 ± 1.15 and 82.46 ± 1.89 (g whey/100 g milk) in the A2 and control milk, respectively.

Water-holding properties have been well-recognized by food technologists among the diversity of functional properties attributed to milk protein products [44]. For acid coagulation products such as yoghurt, whey separation or syneresis is one of the major problems found because of the undesired texture and instability of processing and storage [45]. The formation of a more stable gel network is related directly with the strong ability to retain water (less syneresis), while a weaker gel (less firm) promotes more whey separation and is more sensitive to syneresis [46]. Although differences between milks were not statistically significant, the A2 milk presented lower WHC than the control milk, which would result in a denser gel, aligning with the previous results (Table 4).

Knowledge related to the topic of WHC with respect to β-CN polymorphism in milk is limited to date. Ketto et al. [47] studied the influence of milk casein genotypes on the degree of syneresis in order to use protein genomics for improving the WHC of cultured milk. However, only the genotypes of α_s1_-CN, κ-CN, and β-Lg were studied. These authors found that the β-LG and κ-CN/β-LG composite genotypes significantly influenced the degree of syneresis, being lower in cultured milk with the AB genotype of β-Lg compared to BB.

### 3.4. Rennet Coagulation Properties

Results of rennet coagulation can be seen in Table 6. As in acid coagulation, no statistical differences between the two types of milk were found in clotting time. Several authors described the effects of the individual genetic variants in rennet coagulation properties, showing shorter coagulation time in the β-CN A1A1 variant compared with β-CN A2A2 [4,21,22,28,29,48].

A clear effect regarding milk type was found on curd firming rate (*p* ≤ 0.01) and curd firmness (*p* ≤ 0.001), both being significantly higher in the A2 milk than the control milk (Table 6). The higher curd firmness observed in the A2 milk coincides with the higher gel density described in acid coagulation (Table 4), but is not in line with the results of Hallén et al. [29], Gustavson et al. [21], and Ketto et al. [22], who found higher rennet curd firmness in β-CN A1A2 than A2A2. These differences in gelation properties could be explained by the higher effect of composite genetic variants than single alleles. It was described that milk coagulation traits are affected by β-κ-CN genotypes, with the most favorable coagulation properties correlating with the composite genotypes containing at least one B allele in both loci [4,23,41]. In addition, comparing both casein loci, it seems that κ-CN affects milk coagulation parameters more than β-CN [23]. Many studies have reported that κ-CN B is associated with short gelation time and high gel strength [21,29,32,49] compared with κ-CN A. Meanwhile, κ-CN E is associated with poor coagulation properties [23,25,29], which could explain the higher curd firmness found in our A2 milk compared to the control milk, because only the control samples presented κ-CN E (Table 2). Gustavson et al. [21] also observed that variations in gelation properties can be better explained with composite genetic variants than with single alleles, describing a diminution of the favorable gelation properties in β-CN A1A1 in combination with κ-CN EE.

A clear effect regarding milk type was also found on potential cheese yield (*p* ≤ 0.001) and syneresis of curds (*p* ≤ 0.001), both being significantly higher in A2 than control milk (Table 7). Abeykoon et al. [50] described a negative and positive correlation of enzymatic coagulum yield with β-CN A1 and B, respectively. However, no correlation was observed with β-CN A2.

Perna et al. [51] described a significant effect of casein composite genotype on cheese yield. They described that the highest yield was from the α_s1_-β-κ-CN (BB-A2A2-BB) genotype, and was mainly attributed to κ-CN BB. Mariani et al. [52] and Aleandri et al. [53] reported a clear effect of κ-casein on cheese yield. Although the effect of single alleles were not studied, low fat and protein contents were reported [51,54,55,56] in cheese whey when using milk with κ-CN BB rather than AA. Marziali and Ng-Kwai-Hang [57] reported that β-CN A1A1 milk showed a higher capacity to retain protein in coagulum during cheesemaking than β-CN A2A1, which should result in higher cheese yield. In the present study, rennet gels made with A2 milk expelled more whey than gels from control milk (Table 7). This higher syneresis found in gels from A2 milk could explain the higher rennet gel firmness attributed to this milk (Table 6).

## 4. Conclusions

The most common types of β-CN in cow’s milk are A1 and A2. Currently, there is a tendency to select animals for β-CN A2A2, which leads to a modification of the genetic variety of milk, which has until now been formed mainly by β-CN A1A2. The present study examines the effect of the increased frequency of the β-CN A2 allele in the tank milk on the acid and rennet coagulation properties of milk. It should be taken into consideration that the animal selection for the β-CN A2A2 genotype influenced the genetic variants of other milk proteins, principally κ-CN. It is described that κ-CN E is associated with poor coagulation properties, and this allele is rarely associated with β-CN A2A2. Therefore, the genetic selection towards milk with β-CN A2A2 modifies the genetic frequency of other milk protein variants, which could influence the coagulation properties of milk. Milk containing only β-CN A2 showed higher acid gel density and curd firmness than control milk (which had a mixture of A1 and A2 variants). These results support the higher syneresis of the enzymatic gels and the slightly lower, although not statistically significant, water-holding capacity of acid gels from A2 milk. The acid and enzymatic coagulation times were not affected in the A2 milk compared to the control milk, and no differences were observed in the fermentation curves. In summary, the A2 milk presented similar acid and rennet coagulation properties compared to the control milk, with only slight differences in some technological parameters, making it possible to create dairy products in a similar way with both milks.

## Figures and Tables

**Figure 1 foods-11-03648-f001:**
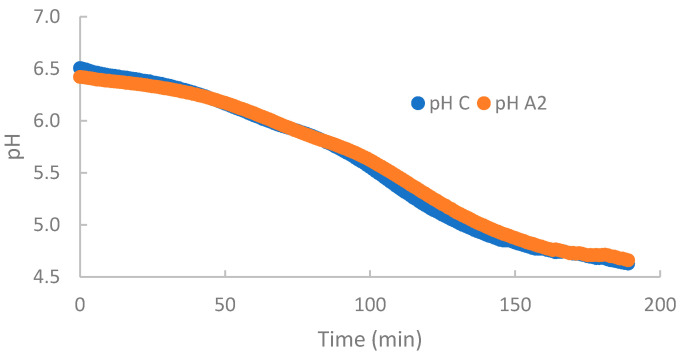
Change in pH over time of A2 and control milks during the acidification process, obtained using a CINAC^®^ device.

**Table 1 foods-11-03648-t001:** Days in lactation and number of lactations of cows used in this study.

Milk ^1^	N ^2^	DIL				Lac			
		Mean	SD	Minimum	Maximum	Mean	SD	Minimum	Maximum
A2	243	167.3	55.3	116.4	272.5	1.52	0.43	1	2.1
C	228	175	48.1	119	265	1.67	0.55	1.1	2.6

^1^ M: Milk samples (A2: milk with A2 β-CN, C: control milk containing β-CN A1 and A2), ^2^ N: Number of cows, DIL: Days in lactation, Lac: Number of lactations.

**Table 2 foods-11-03648-t002:** Genetic information of tank milks used in this study.

		β-CN			κ-CN						β-LG		
Milk ^1^	N ^2^	A2A2	A1A2	A1A1	AA	AB	BB	BE	AE	EE	AA	AB	BB
A2	243	243	0	0	75	117	50				98	108	37
C	228	9	190	29	43	63	28	49	36	5	82	112	36

^1^ Milk samples (A2: milk with A2 β-CN, C: control milk containing β-CN A1 and A2), ^2^ N: Number of cows, β-CN: β-casein, κ-CN: κ-casein, β-LG: β-lactoglobulin.

**Table 3 foods-11-03648-t003:** Chemical composition of milks.

Milk ^1^	pH	TS (%)	Protein (%)
A2	6.67 ± 0.06	9.12 ± 0.25	3.17 ± 0.15
C	6.73 ± 0.05	9.16 ± 0.17	3.09 ± 0.17
* **SE** * ^2^			
Farm			*
Milk type			
Farm × Milk type			

^1^ A2: tank milk with A2 β-CN, C: control tank milk with A1 and A2 β-CN; ^2^
***SE***: statistical significance * *p* ≤ 0.05.

**Table 4 foods-11-03648-t004:** Acid coagulation properties calculated with Optigraph^®^ of A2 and control milks.

Milk ^1^	CT (min)	AR (mA/min)	GD (mA)
A2	108.83 ± 29.38	0.49 ± 0.04	26.66 ± 2.48 ^a^
C	99.66 ± 22.88	0.48 ± 0.03	24.75 ± 3.37 ^b^
* **SE** * ^2^			
Farm		*	
Milk type	*		*
Farm × Milk type		*	*

^1^ A2: tank milk with A2 β-CN, C: control tank milk with A1 and A2 β-CN; ^2^
***SE***: statistical significance * *p* ≤ 0.05. ^ab^ Mean for the same parameter followed by different letters are significantly (*p* ≤ 0.05) different. CT = clotting time, AR = aggregation rate, and GD = gel density.

**Table 5 foods-11-03648-t005:** Acidification properties of A2 and control milks.

Milk ^1^	$V_{max}$ (pHU/min)	$\;V_{m}$ (pHU/min)	$T_{f}$ (min)	$T_{max}$ (min)	$T_{a}$ (min)
A2	0.0192 ± 0.0024	0.0186 ± 0.0016	191.11 ± 15.80	112.67 ± 8.55	23.78 ± 5.61 ^a^
C	0.0200 ± 0.0027	0.0188 ± 0.0028	184.88 ± 21.89	105.63 ± 12.51	12.35 ± 5.83 ^b^
* **SE** * ^2^					
Farm			**		
Milk type					***
Farm × Milk type				*	

^1^ A2: tank milk with A2 β-CN, C: control tank milk with A1 and A2 β-CN; ^2^
***SE***: statistical significance *** *p* ≤ 0.001; ** *p* ≤ 0.01; * *p* ≤ 0.05. ^ab^ Mean for the same parameter followed by different letters are significantly (*p* ≤ 0.05) different. $V_{max}$ = maximum acidification speed; $\;V_{m}$ = mean of acidification rate; $T_{f}$ = time of fermentation; $T_{max}$ = time to reach $\;V_{max}$; $T_{a}$ = latency time.

**Table 6 foods-11-03648-t006:** Rennet coagulation properties calculated with Optigraph^®^ of A2 and control milks.

Milk ^1^	RCT (min)	RCF (mA/min)	CF_30_ (mA)
A2	4.31 ± 2.33	2.48 ± 0.80 ^a^	12.70 ± 1.29 ^a^
C	4.08 ± 2.33	1.89 ± 0.70 ^b^	10.95 ± 0.75 ^b^
* **SE** * ^2^			
Farm			**
Milk type		**	***
Farm × Milk type		**	***

^1^ A2: tank milk with A2 β-CN, C: control tank milk with A1 and A2 β-CN; ^2^
***SE***: statistical significance *** *p* ≤ 0.001; ** *p* ≤ 0.01. ^ab^ Mean for the same parameter followed by different letters are significantly (*p* ≤ 0.05) different. RCT = rennet clotting time, RCF = rate of curd firming, and CF_30_ = curd firmness at 30 min.

**Table 7 foods-11-03648-t007:** Potential yield, total solids and syneresis of A2 and control milks.

Milk ^1^	Potential Yield (%)	TS (%)	Syneresis (g)
A2	17.02 ± 4.77 ^a^	27.53 ± 1.94	9.66 ± 2.21 ^a^
CN	12.89 ± 1.06 ^b^	26.93 ± 0.89	7.26 ± 1.52 ^b^
* **SE** * ^2^			
Farm			
Milk type	***		***
Farm × Milk type	***		***

^1^ A2: tank milk with A2 β-CN, C: control tank milk with A1 and A2 β-CN; ^2^
***SE***: statistical significance *** *p* ≤ 0.001. ^ab^ Mean for the same parameter followed by different letters are significantly (*p* ≤ 0.05) different. TS: total solids.

## Data Availability

The data presented in this study are available upon request from the corresponding author.

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
