# Peer review of "Acid and Rennet Coagulation Properties of A2 Milk"

_foods, 2022, doi:10.3390/foods11223648_

Round 1
Reviewer 1 Report
Manuscript ID: foods-2024130
TITLE: "Acid and rennet coagulation properties of A2 milk" by Bibiana Juan and Antonio-José Trujillo
In this work, the authors evaluated the acid and rennet milk coagulation properties of A2 milk (β-casein (CN) A2A2 genotype) and made a comparison with control milk, which consists in a blend of A2A1/A1A1/A2A2 genotypes. Since there has been a growing interest in A2 milk in recent years, this work may be useful for cheese producers that use this type of milk. However, some inaccuracies in this work needs to be adjusted. They are listed below.
Abstract
Line 8: “This study” is repeated two times
Line 12: “pH=4.6”, delete =
Line 16: the sentence “Acid and enzymatic coagulation times were not affected in milks.” Is better “in both milks”
Materials and Methods
Line 138: Correct the sentence “5000 g for..” in “5000xg for…”
Line 150: Correct the sentence “13.000 g×…” in “13000xg….”
Line 157: Correct “32 ºC” in “32 °C”
Results
Line 190: The units of measure for proteins and total solids-non-fat are missing
Table 3, 4, 5, 6: These tables should be rearranged in a better way…furthermore where is the control (C)? Is it indicated as CN? This is not clear …
Line 258: the WHC value has a unit of measure? In the text is not indicated, please provide it.
My best regards
Author Response
Dear reviewer,
Thank you very much for you revision and your comments.
Abstract
Line 8: “This study” is repeated two times.
We prefer do not introduced the number of repetitions in the abstract. It is well defined in the material and methods. However, if you think that is necessary to introduced it in this place, we will made this change.
Line 12: “pH=4.6”, delete =
Has been corrected
Line 16: the sentence “Acid and enzymatic coagulation times were not affected in milks.” Is better “in both milks”
Has been modified
Materials and Methods
Line 138: Correct the sentence “5000 g for..” in “5000xg for…”
Has been corrected
Line 150: Correct the sentence “13.000 g×…” in “13000xg….”
Has been corrected
Line 157: Correct “32 ºC” in “32 °C”
Has been corrected
Results
Line 190: The units of measure for proteins and total solids-non-fat are missing
Have been added
Table 3, 4, 5, 6: These tables should be rearranged in a better way…furthermore where is the control (C)? Is it indicated as CN? This is not clear …
The tables have been corrected modifying CN to C. However, we don’t understand which arrangement do you need. We think that is correct this presentation, with the statistical results, as is presented in a several papers.
Line 258: the WHC value has a unit of measure? In the text is not indicated, please provide it.
Have been added

Reviewer 2 Report
The topic of the article „Acid and rennet coagulation properties of A2 milk” falls within the thematic scope of FOODS.
The aim of the research was to determine the acid and enzymatic coagulation properties of tank milk obtained from cows with the β-CN A2A2 genotype, in comparison to a control milk (mixed of milks from cows with A1A2, A1A1 and A2A2 β-CN genotypes). The research topic is highly up-to-date, information about the technological usefulness of A2 milk is still too little. Acid and rennet coagulation were evaluated by Optigraph® System, the acidification kinetics was monitored using a CINAC® System. The water-holding capacity of acid milk gels, the potential yield, total solids and syneresis of enzymatic gels, were also evaluated. It is worth emphasizing the choice of such (instrumental) research methods that ensure the objectivity of the results.
I rate the manuscript very highly both in terms of the topic and methodical approach to it. Therefore, all comments are intended only to improve it.
I have only few comments to the manuscript - all suggestions for corrections were introduced in the review mode to the attached pdf file. The main concern - wrong way of citing references in the text - this requires the Authors to carefully review the text and introduce changes.

Author Response
Dear reviewer,
Thank you very much for you revision and your comments. All the suggestion for corrections have been introduced.